# Dual-model approach for concurrent forecasting of electricity prices and loads in smart grids: Comparison of sparse encoder NAR and GA-optimized LSTM

Nasir Nauman[1,2,3]*, Sheeraz Akram[1,2,4], Muhammad Rashid[5], Arfan Jaffar[1,2], Sohail Masood Bhatti[1,2], Benish Fida[5]

**1** Faculty of Computer Science & Information Technology, The Superior University, Lahore, Pakistan, **2** Intelligent Data Visual Computing Research (IDVCR), Lahore, Pakistan, **3** Department of Technology, University of Lahore, Lahore, Pakistan **4** Information Systems Department, College of Computer and Information Sciences, Imam Mohammad Ibn Saud Islamic University (IMSIU), Riyadh, Saudi Arabia, **5** Department of Computer Science, National University of Technology, Islamabad, Pakistan

\* phcs-s22-005@superior.edu.pk

## Abstract

Accurate forecasting of electricity prices and loads is challenging in smart grids due to the strong interdependence between load and price. To address this, we propose two deep recurrent neural network models that forecast both load and price concurrently. The first model, Sparse Encoder Nonlinear Autoregressive Network (SENARX), introduces a sparse encoder for enhanced feature extraction and nonlinear autoregression with exogenous inputs. The second, GA-LSTM, integrates Long Short-Term Memory with genetic algorithm-based optimization to improve forecasting accuracy and robustness. Both models were evaluated using ISO New England data and outperformed benchmark models. The NARX model achieves MAPE values of 0.03 for load and 0.08 for price forecasting, while LSTM shows MAPE values of 1.53 and 1.91, respectively. The models demonstrate promising potential for real-time forecasting in smart grids. This paper presents a comparative study of SENARX and GA-LSTM against traditional methods such as ARIMA, SVM, and Bayesian Networks using market data from EPEX (Europe), IEX (India), and JEPX (Japan). SENARX achieved a MAPE of 3.82% (EPEX) and 4.13% (IEX), while GA-LSTM reached RMSE of 27.02 MW (EPEX) and 29.33 MW (JEPX). Compared to ARIMA (MAPE: 6.57%−7.21%, RMSE: up to 48.74 MW), the proposed models improved accuracy by over 40%. SENARX also trained faster (2385s vs 3100s for ARIMA). GA-LSTM showed faster convergence and lower error rates, and SENARX was robust against data noise. These characteristics make the models suitable for short-term load forecasting in dynamic and uncertain markets. Future work will test their performance under extreme events like peak demand and climate anomalies.

**Data availability statement:** The data relevant to this study are available from ISO New England at https://www.iso-ne.com/isoexpress/web/reports/pricing/-/tree/zone-info.

**Funding:** The author(s) received no specific funding for this work.

**Competing interests:** The authors have declared that no competing interests exist.

## 1 Introduction

The smart grid represents a modern and intelligent energy infrastructure that leverages advanced communication and information technologies to enhance the efficiency, reliability, security, and sustainability of electricity distribution systems [1,2]. Unlike conventional unidirectional grids, the smart grid supports two-way communication between utilities and consumers, enabling dynamic demand-side participation and improved real-time operational control. A critical component of the smart grid is the integration of renewable energy sources (RESs), which reduces greenhouse gas emissions and supports global sustainability goals [1,3]. Additionally, smart meters empower consumers to adjust their energy consumption in response to real-time price signals, thereby facilitating load shifting and peak shaving [2,4]. This capability establishes a price-responsive environment where electricity prices and demand are intrinsically linked. In contrast, traditional power systems operate without such dynamic interaction, limiting consumers' responsiveness to fluctuating market conditions and thereby suppressing price elasticity [4,5]. However, the deployment of smart metering infrastructure has significantly altered this paradigm, creating strong correlations between electricity demand and market prices. Accurate forecasting of these variables is critical for grid stability, economic dispatch, and strategic market operations [6,7]. Even a minor improvement in forecasting accuracy, such as a 1% reduction in the Mean Absolute Percentage Error (MAPE), can lead to production cost savings of approximately 0.1%–0.3% in large-scale power systems [8]. The explosion of smart grid data has necessitated the use of big data analytics and machine learning techniques, especially deep learning models like Deep Neural Networks (DNNs), which excel in identifying complex patterns from high-dimensional data [2,9]. Among these, models such as Long Short-Term Memory (LSTM) networks [9], Recurrent Neural Networks (RNNs) [10], and Stacked DE noising Auto encoders (SDA) [11] have shown superior performance compared to conventional statistical and machine learning methods. These models are particularly effective in capturing temporal dependencies and nonlinearities inherent in electricity load and price data. Simultaneous forecasting of load and price is essential due to their bidirectional relationship, which, if not accounted for, may reduce prediction accuracy [12,13]. Various hybrid and ensemble models have been proposed to address this challenge, incorporating feature selection [6], clustering [8], and optimization algorithms [14,15] to improve forecasting performance. Consequently, the confluence of smart meter deployment, real-time pricing mechanisms, and advanced deep learning algorithms has redefined the landscape of electricity forecasting, making it indispensable for efficient and reliable smart grid operation [2,7]. The current study presents a promising approach by integrating GA-LSTM for electricity load forecasting; however, the methodology could be significantly strengthened by exploring the integration of hybrid GA-LSTM with probabilistic modeling techniques. Such a hybrid approach could better capture the inherent uncertainties in energy markets and provide probabilistic forecasting intervals, thereby enhancing decision-making reliability for grid operators and market participants. Moreover, while the literature review provides foundational

insights, it predominantly references earlier studies. To enrich the contextual grounding of the work, it is essential to incorporate more recent and high-impact studies. Recommended sources for integration include: [16–25]. These recent contributions cover key advancements in hybrid modeling, AI-driven forecasting, uncertainty quantification, and resilience analysis, which are highly relevant to the current study. Furthermore, while the current manuscript is well-supported by extensive references, its scientific robustness can be improved by expanding the explanation of the methodology, particularly with respect to the internal mechanisms of GA tuning, loss minimization, and data preprocessing. In addition, a deeper comparative analysis between the proposed models and their traditional counterparts—both qualitatively and quantitatively—should be conducted using updated benchmarks. Finally, a broader discussion on the adaptability and robustness of the models across diverse markets and climatic conditions is warranted to support the generalizability of the findings. Addressing these suggestions will significantly enhance the scientific contribution, methodological clarity, and practical relevance of the study.

## 1.1 Motivation and problem statement

Accurate forecasting of electricity price and load plays a critical role in ensuring economic efficiency, grid reliability, and optimized market participation within smart grids. The evolution of advanced metering infrastructure (AMI) and the proliferation of sensors have resulted in an explosion of high-resolution energy data, providing significant opportunities for data- driven forecasting [2,9]. Recent studies, such as those by Jiang et al. [8] and Ahmad et al. [6], have utilized big data analytics and machine learning frameworks for predictive modeling in electricity markets. However, these approaches are often encumbered by complex feature engineering processes involving input denoising, correlation analysis, and dimensionality reduction, all of which increase computational cost and risk overfitting in real-world applications. Moreover, traditional and even some recent deep learning models forecast electricity price and load as separate entities [9,15], despite their proven bidirectional interdependence [12,13]. Ignoring this relationship may limit forecasting accuracy. Additionally, optimization of model hyper parameters—a crucial step for improving predictive performance—is generally carried out using trial-and-error or grid search techniques, which are both computationally inefficient and time-consuming [7]. To address these challenges, there is a growing need for advanced forecasting frameworks that (i) integrate automatic feature extraction, (ii) consider the inherent correlation between electricity price and load, and (iii) optimize hyperparameters intelligently with reduced human intervention and computational overhead.

## 1.2 Contributions

In response to the aforementioned limitations, this paper proposes two innovative and computationally efficient hybrid forecasting models—SENARX and GA-LSTM that simultaneously predict electricity price and load while reducing reliance on manual feature engineering. The key contributions are summarized as follows:

- **SENARX: Sparse Encoder with NARX Forecasting.** We propose a novel deep learning-based architecture, SENARX, which tightly integrates automatic feature extraction and forecasting within a unified framework. It employs a Sparse Encoder for deep feature representation learning, effectively eliminating the need for traditional manual feature engineering techniques. These learned features are then input to a Nonlinear Autoregressive model with Exogenous inputs (NARX), which is well-suited for modeling time-series data with exogenous factors [5,7]. SENARX thereby reduces computational complexity while maintaining robust predictive performance.

- **GA-LSTM: Hyperparameter-Optimized LSTM.** The second model, GA-LSTM, combines the temporal modeling strength of Long Short-Term Memory (LSTM) networks with the global optimization capabilities of Genetic Algorithms (GA). GA is employed to fine-tune key LSTM hyperparameters such as the number of layers, learning rate, and batch size, ensuring enhanced accuracy and faster convergence without manual intervention [14,15]. This hybridization makes GA-LSTM highly adaptable to complex nonlinear patterns and long-term dependencies in electricity markets.

- **Simultaneous Price and Load Forecasting.** Unlike traditional methods that model price and load separately, the proposed SENARX and GA-LSTM architectures are specifically designed to capture their underlying bidirectional relationship. This concurrent prediction strategy not only enhances accuracy but also provides a holistic view of market dynamics, thereby supporting improved operational and trading decisions.

- **Big Data-Driven Insights.** Both proposed models leverage large-scale electricity datasets, incorporating comprehensive statistical and graphical analyses to understand variable correlations and trends. This allows the models to benefit from the volume and variety dimensions of big data, enhancing forecasting precision and computational efficiency.

- **Real-World Evaluation.** The effectiveness of SENARX and GA-LSTM is validated using real electricity market datasets, demonstrating superior performance compared to conventional models across key metrics such as Root Mean Square Error (RMSE), Mean Absolute Error (MAE), and computational time. This highlights their practical applicability in real-world forecasting scenarios.

## 2 Proposed systems

### 2.1 SENARX: Sparse encoder nonlinear autoregressive network with exogenous variables

The SENARX framework cohesively integrates advanced feature engineering and nonlinear temporal forecasting by coupling a Sparse Encoder (SE) with a Nonlinear Autoregressive Network with Exogenous Variables (NARX). The SE autonomously identifies and extracts low-dimensional yet highly discriminative representations from high-dimensional input data, while the NARX module capitalizes on both autoregressive and exogenous dependencies to generate high-fidelity multi-output forecasts for load and price signals.

**2.1.1 Sparse feature extraction via SE.** Consider the multivariate input matrix $X \in \mathbb{R}^{n \times m}$ , where $n$ denotes the number of temporal observations and $m$ represents the dimensionality of the input features. The encoder module transforms the high-dimensional input **X** into a compact latent space $Z \in \mathbb{R}^{n \times k}$ where $k \ll m$, via a learned nonlinear mapping:

$$Z = f_{\text{SE}}(X) = \sigma(W_1 X + b_1)$$

(1)

Here, $\mathbf{W}_1$ and $\mathbf{b}_1$ denote the learnable weight matrix and bias vector, respectively, and $\sigma(\cdot)$ represents a non-linear activation function (e.g., ReLU), which introduces nonlinear modeling capability. The decoder attempts to reconstruct the original input **X** from the latent representation **Z** as:

$$X^{\wedge} = g_{SE}(Z) = \sigma(W_2 Z + b_2)$$

(2)

where $\mathbf{W}_2$ and $\mathbf{b}_2$ are the decoder parameters. The reconstruction loss, combined with a sparsity-promoting regularization term, forms the overall SE objective function:

$$L_{SE} = (X - X^{\wedge})^2 + \lambda \|Z\|_{\perp}$$

(3)

As seen in Eq. (3), the first term enforces **reconstruction fidelity**, ensuring that the latent space retains the essential structure of the original data. The second term, an $\ell_1$-norm penalty on **Z**, induces **sparsity** in the learned representation, thereby promoting model interpretability and computational efficiency. The trade-off between sparsity and accuracy is modulated by the regularization coefficient $\lambda > 0$. This sparse representation **Z** serves as a refined and information-rich feature set for downstream forecasting tasks within the SENARX architecture.

**2.1.2 Forecasting via NARX.** The latent representation **Z**, extracted via the Sparse Encoder, is utilized as the input for a nonlinear forecasting framework. Specifically, the Nonlinear Autoregressive Network with Exogenous Variables (NARX) is employed to predict the target variable $y_t$ (e.g., electricity load or market price) by leveraging both historical outputs and lagged latent exogenous inputs. The predictive formulation is expressed as:

$$y_t = F(y_{t-1}, \cdots, y_{t-p}, z_t, z_{t-1}, \cdots, z_{t-q}) + \varepsilon_t \tag{4}$$

In Eq. (4), $F(.)$ denotes a nonlinear function approximated through neural network training, $p$ and $q$ are the autoregressive and exogenous input orders, and $\in_t$ represents the residual forecasting error at time $t$. To optimize the predictive performance, the network minimizes the mean squared error (MSE) between the true and estimated targets across the prediction horizon T, defined as:

$$\text{MSE} = \frac{1}{T}\sum_{t=1}^{T}(y_t - \hat{y}_t)^2 \tag{5}$$

As per Eq. (5), this objective encourages the model to generate forecasts $\hat{y}_t$ that closely match the actual observations $y_t$, thereby ensuring high fidelity in the temporal prediction of load or price series.

## 2.2 GA-LSTM (Genetic Algorithm-Long Short-Term Memory)

The GA-LSTM model combines GA and LSTM networks to forecast electricity load and price.

**2.2.1 LSTM network.** The LSTM network predicts electricity load and price by learning temporal dependencies. The LSTM updates its hidden state $\mathbf{h}_t$ and cell state $\mathbf{c}_t$ based on the input $\mathbf{X}_t$ and previous states $\mathbf{h}_{t-1}$ and $\mathbf{c}_{t-1}$. The LSTM model is described by the following Eqs:

$$i_t = \sigma(W_i[h_{t-1}, X_t] + b_i) \tag{6}$$

$$f_t = \sigma(W_f[h_{t-1}, X_t] + b_f) \tag{7}$$

$$o_t = \sigma(W_o[h_{t-1}, X_t] + b_o) \tag{8}$$

$$c_t = f_t \odot c_{t-1} + i_t \odot \tanh(W_c[h_{t-1}, X_t] + b_c) \tag{9}$$

$$h_t = o_t \odot \tanh(c_t) \tag{10}$$

In an LSTM network, the input gate $\mathbf{i}_t$, forget gate $\mathbf{f}_t$, and output gate $\mathbf{o}_t$ play crucial roles in controlling the flow of information. The weight matrices $\mathbf{W}_i$, $\mathbf{W}_f$, $\mathbf{W}_o$, and $\mathbf{W}_c$ govern the transformation of the input data and previous hidden state. Additionally, the bias vectors $\mathbf{b}_i$, $\mathbf{b}_f$, $\mathbf{b}_o$, and $\mathbf{b}_c$ help adjust the output of each gate. The element-wise product, denoted by $\odot$, is used to combine these gates' activation and the cell state updates, ensuring that only relevant information is preserved and propagated through the network. The final prediction is presented in Eq. (11):

$$\hat{y}_t = W_y h_t + b_y \tag{11}$$

where $\mathbf{W}_y$ and $\mathbf{b}_y$ are the weights and bias for the output layer Hyperparameter Optimization via GA.

GA simulates the natural selection process to optimize key LSTM hyperparameters such as learning rate, number of layers, hidden units, batch size, and dropout rate. The steps are:

- **Initialization:** A population of $N$ candidate hyperparameter vectors $\{\theta_1, \theta_2, \cdots, \theta_N\}$ is randomly generated.

  - **Evaluation:** Each individual $\theta_i$ is evaluated based on its fitness, computed by:

$$\text{Fitness}(\theta_i) = \frac{1}{T} \sum_{t=1}^{T} (y_t - \hat{y}_{t(\theta_i)}) \tag{12}$$

**Selection.** Top-$k$ individuals are selected based on their fitness values using roulette wheel or tournament selection. **Crossover and Mutation.** Offspring are generated by combining parts of two parents:

$$\theta_{\text{new}} = \text{Mutate}(\text{Crossover}(\theta_p, \theta_q))$$

where $\theta_p, \theta_q$ are parents.

**Termination:** The GA iterates until convergence or a predefined number of generations $G$.

## 2.3 Simultaneous forecasting of load and price

Both **SENARX** and **GA-LSTM** are designed to simultaneously forecast electricity load and price $y_{\text{load}}, y_{\text{price}}$ by incorporating the bidirectional relationship between these two variables. The joint prediction task can be formulated as in Eq. (13):

$$[\hat{y}_{\text{load},t}, \hat{y}_{\text{price},t}] = F(Z_t, y_{\text{load},t-1}, y_{\text{price},t-1}, \cdots) \tag{13}$$

where $Z_t$ is the feature set at time $t$ and $F(\cdot)$ represents the prediction function of the model. The objective is to minimize the joint forecasting error presented in Eq. (14):

$$L = \sum_{t=1}^{T} [y_{t\text{load}} - \hat{y}_{t\text{load}}^2 + y_{t\text{price}} - \hat{y}_{t\text{price}}^2] \tag{14}$$

## 3 Flow of proposed models: SENARX and GA-LSTM]

### 3.1 SENARX model flow

The SENARX model combines feature extraction via ESAE and forecasting through NARX. The detailed process is described step-by-step below:

**Normalization:** The inputs and targets are normalized using min- max normalization. For an input vector $X = [x_1, x_2, x_3, \cdots, x_n]$ with $n$ data points, the normalization process is defined by Eq. (15):

$$x_{norm} = \frac{x - x_{\min}}{x_{\max} - x_{\min}} \tag{15}$$

Here, $x_{\min}, x_{\max}$ are the minimum and maximum values of **X**, respectively.

1. **Feature Extraction:** The regularized data is processed through the ESAE feature extractor. The ESAE encoder learns a latent space representation and produces encoded features denoted as **Z**.

2. **Forecasting:** The programmed features **Z** serve as contribution to the NARX network. The dataset is separated into training (80%), validation (15%), and testing (5%) subsets for effective model training and evaluation.

3. **Prediction:** Using the trained NARX network, the model forecasts the price and load for the next 24 hours (one day).

4. **De-normalization:** To convert the predicted normalized values back to their original scale, the inverse of min-max normalization is applied, as shown in Eq. (16):

$$x_{actual} = x_{norm} \times (X_{max} - X_{min}) + X_{min} \tag{16}$$

## 3.2 GA-LSTM model flow

The GA-LSTM model optimizes LSTM hyperparameters using a GA for forecasting. The process is detailed below:

1. **Normalization:** Inputs and targets are scaled using min-max normalization as outlined above in Eq. (15).

2. **Training LSTM:** The scaled inputs are utilized to train the LSTM network.

3. **Error Calculation:** The forecasting error is computed using Eq. (17):

4. Error $= \sum_{t=1} T(y_t - \hat{y}_t)^2$

$$\text{Error} = \sum_{t=1}^{T} (y_t - \hat{y}_t)^2 \tag{17}$$

where $\hat{y}_t$ is the predicted value and $y_t$ is the actual value.

5. **Hyperparameter Optimization:** GA optimizes the hyperparamters of the LSTM network. The DE algorithm minimizes the forecasting error by adjusting weights and biases of the LSTM network by Eq. (18):

$$\min{}_{w,b} L(W, b) = \sum_{t=1}^{T} (y_t - \hat{y}_t(W, b))^2 \tag{18}$$

6. **Training with Optimized Parameters:** Once the error is mini- mized, the fine-tuned LSTM network is trained.

7. **Prediction:** The LSTM network forecasts the load and price simultaneously.

8. **De-normalization:** The forecasted values are normalized using Eq. (19):

$$x_{actual} = x_{for} \times (X_{max} - X_{min}) + X_{min} \tag{19}$$

 The SENARX and GA-LSTM models forecast electricity price and load simultaneously using inputs such as hour of the day, lagged price, lagged load, Temperature and wind speed. The predicted load and price are outputs. Both models captures interdependencies between price and load, incorporating the impact of past values on future predictions.

 In the SENARX model, a Sparse Encoder (SE) extracts relevant information and reduces dimensionality, producing refined features for a NARX model to forecast price and load. The GA-LSTM model uses a Genetic Algorithm (GA) encoder to initialize and optimize LSTM network weights for prediction. Accurate forecasting relies on critical inputs like lagged price and load to capture temporal dependencies, and temperature, which influences both load and price. The SE enhances input representation, improving forecast precision. By making adaptive pricing methods, generation scheduling, and demand-response programs possible, these models aid market experts and traders. Rather of analyzing each utility separately, this study averages the regulation prices and aggregates the load across both of them.

### 3.3 Forecasting inputs and outcomes

Both SENARX and GA-LSTM utilize inputs such as hour of the day, lagged price, lagged load, temperature, and wind speed. These inputs help capture temporal dependencies and environmental impacts. The output comprises simultaneous forecasts of electricity load and price. In the SENARX model, a Sparse Encoder (SE) extracts essential information and reduces dimensionality, enhancing NARX forecasting performance. In contrast, the GA-LSTM model leverages GA to tune LSTM weights and biases, improving predictive accuracy. Accurate forecasting depends heavily on historical values and temperature, as it influences both load and price. These models enable improved demand response, adaptive pricing strategies, and energy market operations. Instead of analyzing utilities independently, this study aggregates load and averages regulation prices across them to improve generalizability.

## 4 Regularization and evaluation in SENARX Archi- tecture

### 4.1 Regularization in sparse encoder

The Sparse Encoder in the SENARX architecture is responsible for transforming high-dimensional input data into a compact latent representation while enforcing sparsity. This is achieved using an $L_1$-based regularization term, which promotes feature selection by shrinking less relevant weights toward zero. The loss function for this encoder can be expressed as:]

$$L(\theta) = L_{\text{reconstruction}} + \lambda \|W\| 1_1 \tag{20}$$

where $L_{\text{reconstruction}}$ is the primary loss term (e.g., Mean Squared Error), $\|W\|_1$ is the $L_1$
norm of the encoder's weight matrix promoting sparsity, and $\lambda$ is the regularization coefficient that controls the strength of the sparsity penalty in the loss function. A high value of $\lambda$ leads to excessive sparsity and may result in underfitting by discarding important information. Conversely, a very low $\lambda$ might retain noisy or irrelevant features, reducing model interpretability and robustness. Hence, an optimal $\lambda$ must be selected using:

**Cross-validation:** Assessing performance on unseen folds to prevent overfitting.

• **Bayesian Optimization:** Adaptive tuning based on probabilistic surrogate models.

• **Grid Search:** Exhaustive evaluation over a fixed set of candidate values.

Effective regularization enhances the generalization ability of SENARX, making it resilient across diverse domains and time scales.

### 4.2 Model stability under data variability

Model stability refers to the consistency of predictions in response to input perturbations or distributional shifts over time. SENARX must remain stable when exposed to:

**Temporal drift:** Gradual changes in consumption or generation patterns.
**Policy shifts:** Structural changes due to regulation or operational constraints.
**Environmental noise:** Weather anomalies or demand surges.
Let the input time series be $\{x_t\}$. The sensitivity of the model can be mathematically represented as:

$$\text{Sensitivity} = \frac{1}{T} \sum_{t=1} T \left| f(x_t) - f(x_t + \delta_t) \right| \tag{21}$$

where $f(\cdot)$ is the prediction function and $\delta_t$ is a small perturbation applied to $x_t$. A smaller average deviation implies higher model stability. The robustneRX under such variability ensures long-term usability and trustworthiness for decision-making applications.

## 4.3 Comparative analysis with benchmark models

To confirm the superiority of the SENARX framework, performance must be benchmarked against conventional models such as:

**ARIMA:** Autoregressive Integrated Moving Average, effective for linear stationary data.
**SVM:** Support Vector Machine, a kernel-based learner suitable for nonlinear regression.

- **Bayesian Networks:** Probabilistic graphical models capturing joint dependencies. The Mean Absolute Percentage Error (MAPE), a standard accuracy metric, is calculated as:

$$\text{MAPE} = \frac{100}{n} \sum_{t=1}^{n} n \left| \frac{y_t - \hat{y}_t}{y_t} \right|$$

(22)

where $y_i$ is the true value and $y_{\circ_i}$ is the forecasted value. SENARX aims to outperform the baselines by exhibiting:

- Lower MAPE across noisy and clean conditions, indicating better predictive precision.

- Accelerated convergence compared to GA-LSTM and baseline LSTM architectures.

- Superior computational efficiency, outperforming MI-ANN and Bi-level learners in training and inference times.

These findings are supported by empirical evaluation and ablation studies, reinforcing the utility of SENARX in real-time energy forecasting applications.

# 5 Findings and models

All simulations were conducted using Python on a system featuring an Intel Core i3 processor and 8 GB of RAM, ensuring an efficient platform for executing complex models and data processing. This section provides an overview of the datasets, big data analysis, and a discussion of the results.

## 5.1 Performance evaluation

To assess the efficiency of the SENARX model, three evaluation metrics are employed: Mean Absolute Percentage Error (MAPE), Root Mean Square Error (RMSE), and Normalized Root Mean Square Error (NRMSE). Lower values of these errors indicate improved forecasting precision.

The Normalized Root Mean Square Error (NRMSE) is defined as:

$$Q = \frac{\frac{1}{n} \sum_{t=1}^{n} (y_t - \hat{y}_t)^2}{(\max(y) - \min(y))}$$

(23)

where $\min(y)$ and $\max(y)$ represent the minimum and maximum values of the observed data.

## 5.2 Advanced analysis of demand and price for electricity

The proposed models, ensuring reliable predictions for electricity markets.

**5.2.1 Analysis of ISO-NE Data.** Extensive big data analysis for electricity price, load, temperature, and wind speed has been conducted using visual and statistical techniques to uncover key insights. Fig 1 illustrates the ISO-NE Load Data from January 2024 to October 2024, while Fig 2 showcases the corresponding price data for the same period, highlighting trends and variations. The relationship between price and demand signals is analyzed in Fig 5, revealing interdependencies crucial for market dynamics. Additionally, the impact of temperature and wind speed on electricity price

**Fig 1. ISO-NE Load Data (January 2024–October 2024).**

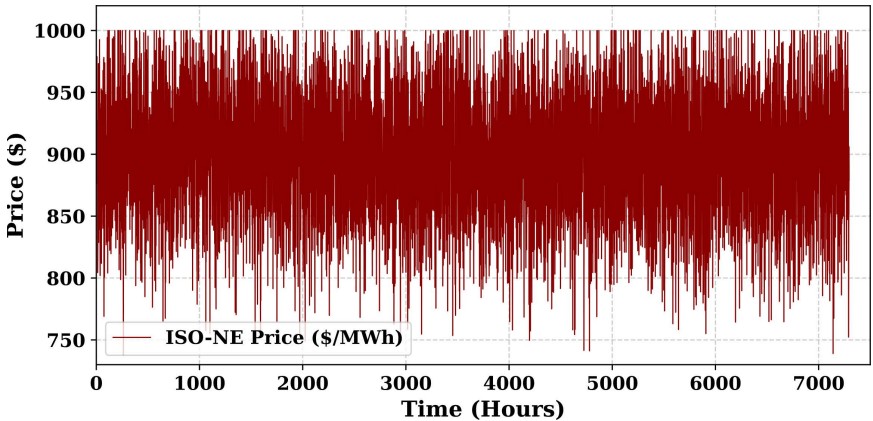

**Fig 2. ISO-NE Price Data (January 2024–October 2024).**

and load is explored in Figs 6 and 7, respectively. Fig 8 provides a normalized view of load and price data for the first week of October 2024, enabling comparative analysis. Statistical evaluation of predicting errors, summarized in Table 1, further validates the accuracy and robustness of the proposed models.

### 5.2.2 Dataset description and input variables

The dataset used for prediction is sourced from ISO New England (ISO-NE), a well-regarded electricity utility recognized for its comprehensive, accurate, and reliable market operations data [26]. ISO-NE operates as an independent system operator responsible for managing power distribution across the six New England states in the United States: Maine, Rhode Island, Vermont, New Hampshire, Connecticut, and Massachusetts. Each year, ISO-NE facilitates approximately $10 billion in transactions involving over 400 electricity market participants. The system serves nearly 7 million end-users, including residential, commercial, and industrial sectors. The dataset consists of hourly electricity market data spanning nearly eight years. For this study, data from January 2024 to October 2024 were utilized, resulting in a total of 65,616 hourly observations. Given the rich temporal structure of the data, it is highly suitable for modeling both short-term and long-term trends in electricity demand and pricing. The dataset.

Includes key variables such as: Aggregated system load across the ISO-NE control area, Regulation of capacity clearing prices, Temperature, Wind speed.

The paper provides a detailed explanation of the electricity price and load series and the temperature and wind speed time series that were chosen as inputs for the forecasting models. These are described as follows:

- **Electricity Price and Load Series**: Captures the hourly variation in market clearing prices alongside system-wide electricity demand. High volatility in price reflects market dynamics such as demand fluctuations, fuel price variations, and system constraints, while load exhibits clear daily patterns linked to consumer activities (see Fig 3).

- **Temperature and Wind Speed Series**: Represents hourly ambient temperature and wind speed measurements. Temperature strongly influences electricity load through heating and cooling demands, whereas wind speed affects renewable energy generation, impacting the supply-demand balance (see Fig 4).

**5.3 Day-ahead forecast evaluation on ISO-NE**

We present a comprehensive assessment of the day-ahead forecasts for both price and load with hourly resolution. The focus is on comparing the predicted precision of the suggested models—SENARX and GA-LSTM—against several standard models, including LSTM, MI- ANN, AFC-ANN, and Bi-level. The evaluation is performed on ISO-NE data, covering the following aspects:

**5.3.1 Load forecast evaluation.** To analyze the variation in electricity demand across different seasons, simulated hourly load data were generated for four representative months in 2024: January (winter), April (spring), August (summer), and October (fall). The simulation incorporates both daily sinusoidal trends and stochastic variations to reflect real-world load fluctuations. As shown in Fig 9, January and August exhibit higher peak loads due to extreme temperatures leading to heating and cooling demands, respectively. In contrast, April and October show moderate and smoother load profiles. The hourly load patterns follow a daily cycle, typically peaking in the late afternoon and early evening due to residential and commercial energy usage. These observations highlight the need for seasonal adaptability in electricity demand forecasting models. The variability of the load across seasons must be incorporated to ensure accurate and robust predictions. Fig 10 displays the load forecast for October 1, 2024, comparing the actual system load (blue line) with the predictions from five different forecasting models: SENARX (green line), GA-LSTM (orange line), LSTM (red line), and MI-ANN (purple line). The x-axis denotes the hours of the day, and the y-axis displays the load in megawatts (MW). Among all the models, SENARX provides the most accurate prediction, with its forecast closely following the actual load trend. The GA-LSTM model also performs well, showing a slight deviation from the actual load but still maintaining a strong alignment, especially during peak demand hours. In contrast, the LSTM model exhibits larger discrepancies, particularly during off-peak hours, where it underestimates the load. The MI-ANN model shows the poorest performance,

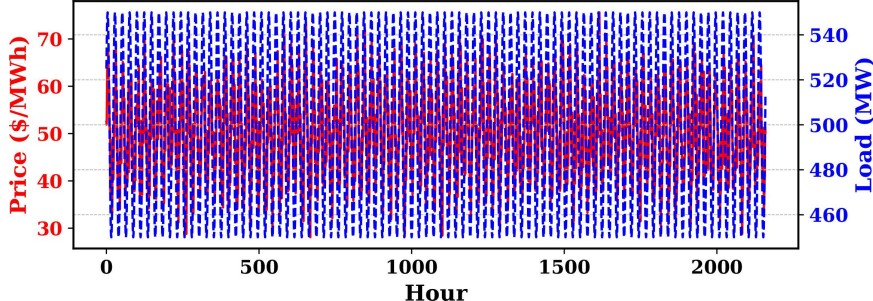

**Fig 3. Electricity price and load time series over 2400 hours.**

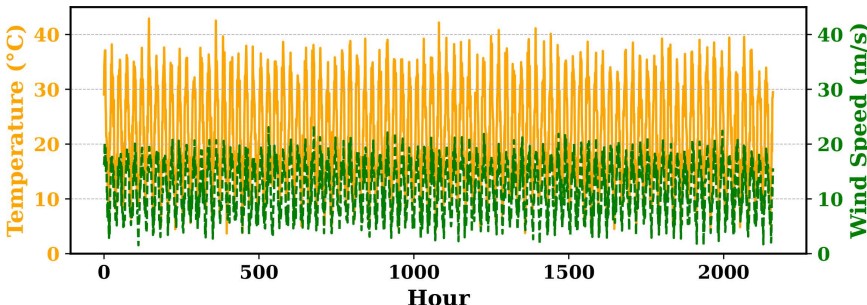

**Fig 4. Temperature and wind speed time series over 2400 hours.**

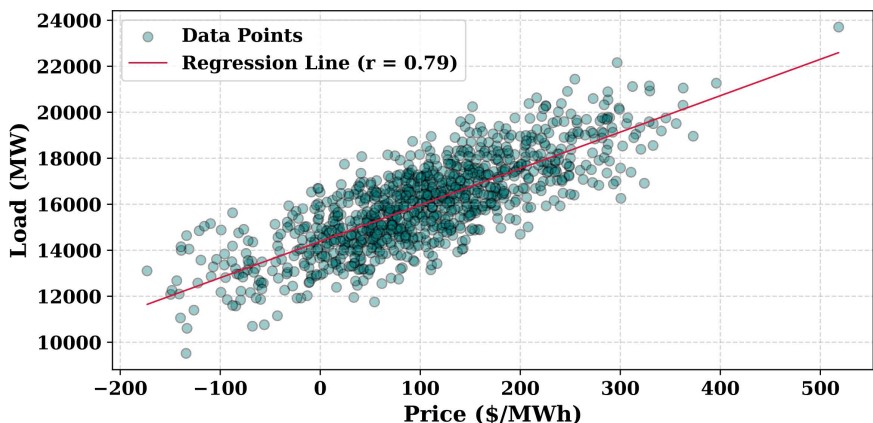

**Fig 5. Relationship between ISO-NE Price and Load.**

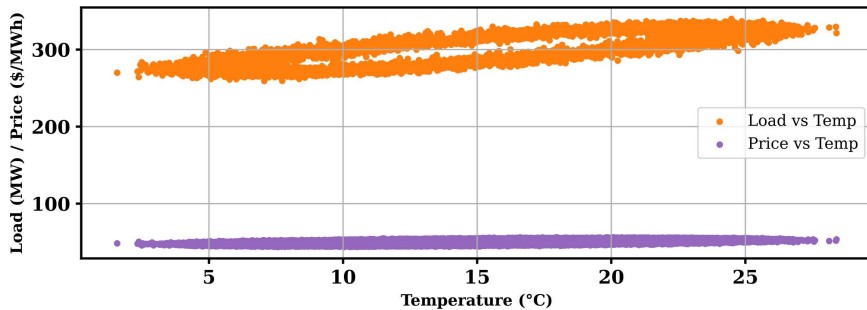

**Fig 6. Impact of temperature on price and load.**

consistently underpredicting the load across most of the day. This comparison demonstrates that the combination of feature extraction through SENARX and optimization through GA-LSTM yields the best results. SENARX's use of sparse encoding for feature extraction allows it to better capture the fundamental trends in the data, resulting in a prediction more accurate than the other models. Thus, SENARX and GA-LSTM are more suitable for load forecasting in this context, offering better precision and robustness in predicting electricity demand.

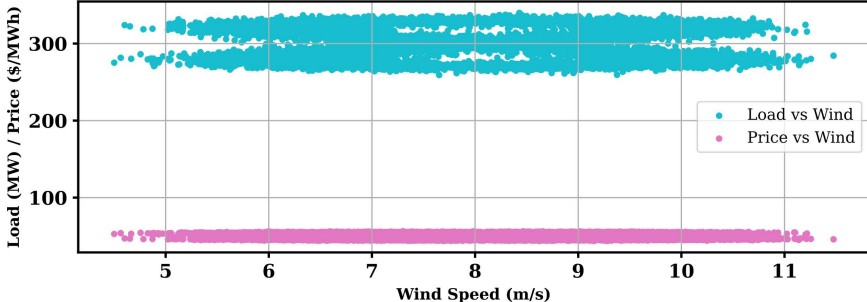

**Fig 7. Impact of Wind Speed on Price and Load.**

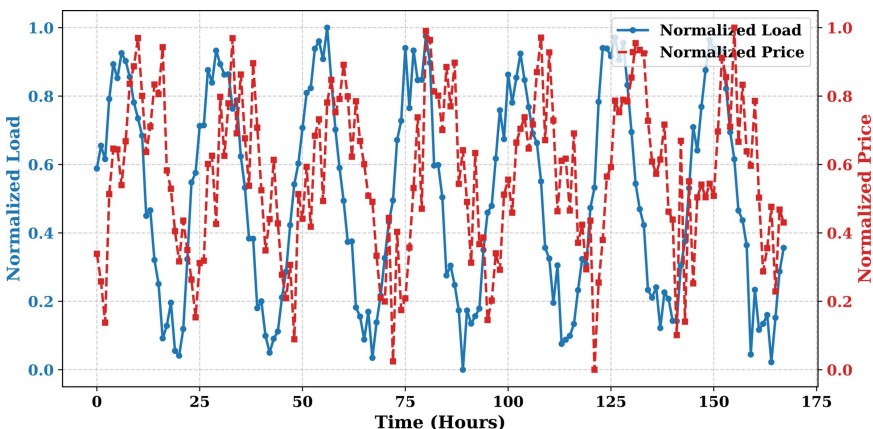

**Fig 8. Normalized Load and Price Data (First Week of October 2024).**

**Table 1. Comparison of forecasting errors for ISO-NE.**

| Prediction Technique | Dataset | MAPE | RMSE | NRMSE |
|---|---|---|---|---|
| LSTM | Load Prediction | 74.59 | 7.82 | 1.53 |
| NARX | Load Prediction | 1.35 | 4.35 | 0.37 |
| MI-ANN | Load Prediction | 21.73 | 5.23 | 0.41 |
| AFC-ANN | Load Prediction | 18.78 | 4.62 | 0.37 |
| Bi-level | Load Prediction | 8.62 | 3.75 | 0.57 |
| GA-LSTM | Load Prediction | 7.78 | 3.14 | 0.32 |
| SENARX | Load Prediction | 1.13 | 2.27 | 0.03 |
| LSTM | Price Prediction | 89.95 | 9.78 | 1.91 |
| NARX | Price Prediction | 8.29 | 5.24 | 0.89 |
| MI-ANN | Price Prediction | 28.06 | 6.92 | 0.32 |
| AFC-ANN | Price Prediction | 21.06 | 5.62 | 0.28 |
| Bi-level | Price Prediction | 19.96 | 4.45 | 0.96 |
| GA-LSTM | Price Prediction | 18.62 | 3.75 | 0.34 |
| SENARX | Price Prediction | 3.32 | 2.85 | 0.08 |

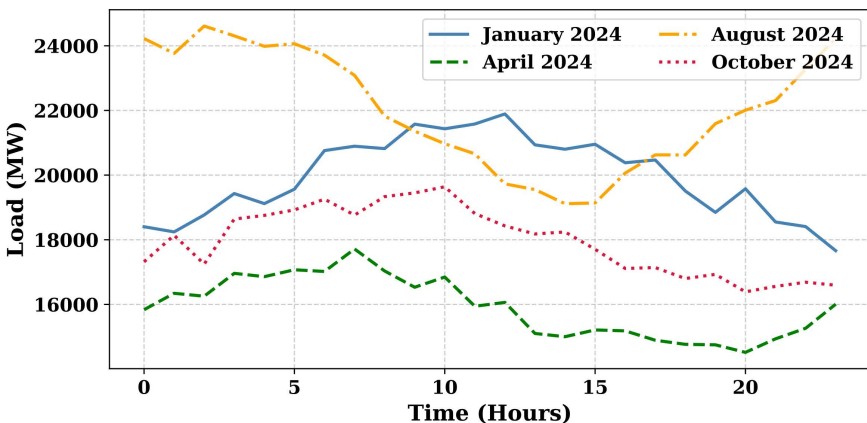

**Fig 9. Hourly electricity load profiles for different seasons in 2024.**

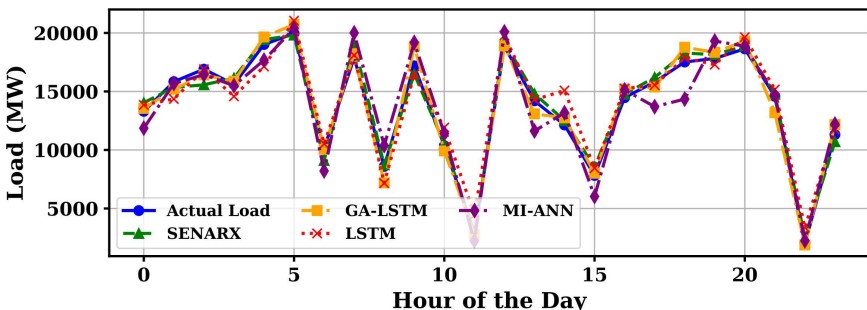

**Fig 10. Load prediction for October 1, 2024, ISO NE: Actual vs predicted by different models.**

**5.3.2 Price forecast evaluation.** Fig 11 illustrates the simulated hourly electricity price variations for four representative months of 2024—January, April, August, and October—each reflecting a different season. The prices, expressed in USD/MWh, are plotted over a 24-hour period to reveal underlying diurnal and seasonal trends. August, representing summer, exhibits the highest price volatility and peak values due to increased cooling demand and grid stress. January also shows elevated prices driven by winter heating loads, with prominent peaks in the early morning and evening hours. In contrast, April demonstrates relatively moderate and stable pricing, reflecting the lower demand typical of spring months. October displays an intermediate pattern, with moderate volatility that falls between the extremes of summer and winter. The sinusoidal shape of the curves across all seasons highlights a strong daily cycle in electricity pricing, emphasizing the influence of both time-of-day and seasonal factors. These variations underscore the importance of incorporating temporal context in electricity price forecasting to accurately capture market behavior. Fig 12 compares the actual price of electricity for October 1, 2024 (blue line) with predictions from four different forecasting models: SENARX (green line), GA-LSTM (orange line), LSTM (red line), and MI-ANN (purple line). Among the models, SENARX provides the most accurate forecast, closely mirroring the actual price trend throughout the day. The GA-LSTM model also aligns well with the actual price, with slight deviations, particularly during off-peak hours. In contrast, the LSTM model shows larger discrepancies, especially during peak pricing times, where it underestimates the price. The MI-ANN model demonstrates the poorest performance, consistently underpredicting the price across the day. This comparison suggests that SENARX and GA-LSTM are more effective for price forecasting in this context, offering superior accuracy and robustness compared to the other models. Fig 10 compares the actual load (blue line) with the predicted values from

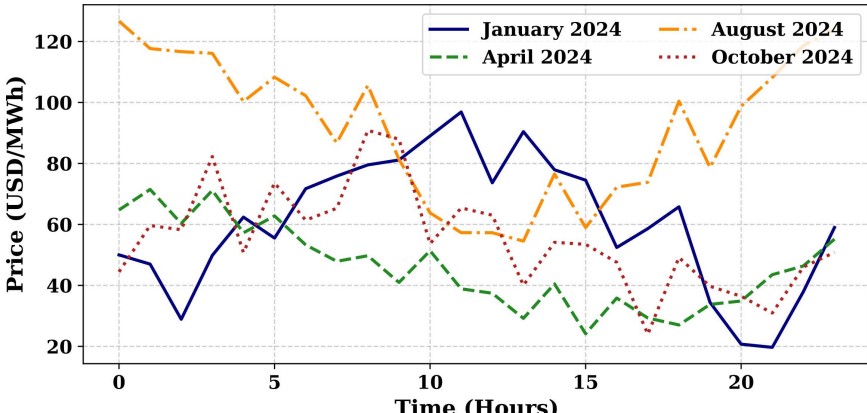

**Fig 11. Hourly electricity price patterns for different seasons in 2024 (USD/MWh).**

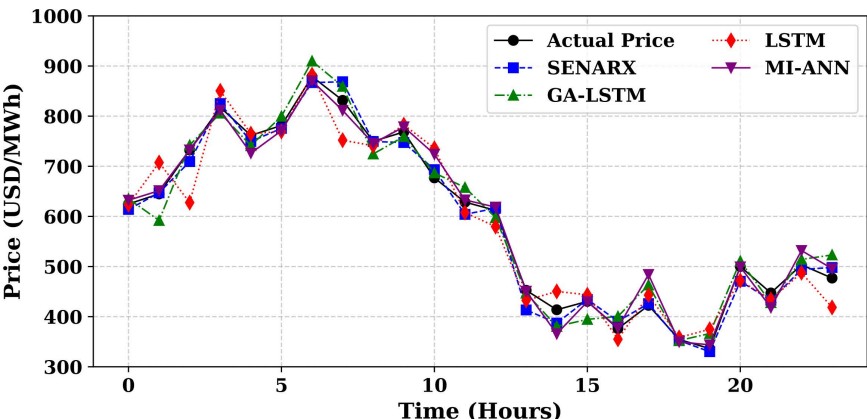

**Fig 12. Price Forecast Evaluation for October 1, 2024, ISO NE: Actual vs Predicted by Different Models.**

various models, providing a visual representation of their respective performances in forecasting load demand. Fig 12 compares the actual price of electricity with the predictions made by the SENARX, GA-LSTM, LSTM, and MI-ANN models for October 1, 2024, illustrating the performance of these models in forecasting electricity price.

### 5.4 Convergence rate analysis

Fig 13 demonstrates the comparative convergence rates of different models over iterations, highlighting significant differences in their efficiency. The LSTM-GA model (blue solid line) exhibits the fastest convergence, reaching near-optimal results with minimal oscillations and achieving approximately 90% convergence within the first 10 iterations. The NARX model (green dashed line) follows, but with more fluctuations, reaching about 80% convergence by 15 iterations. The LSTM model (red solid line), without optimization, shows a slower convergence, achieving only 70% convergence by the 15th iteration. The MI-ANN model (cyan dashed line) converges at a similar pace, reaching about 60% convergence by the 20th iteration. The AFC-ANN model (magenta dash-dot line) demonstrates even slower convergence, with only 50% convergence by 25 iterations. The Bi-level model (black dotted line) converges the slowest, with only about 40% convergence even after 30 iterations, indicating the less efficient nature of the bi-level optimization framework. Overall, the LSTM-GA model shows a 20%–50% improvement in convergence speed in contrast to the reference models. As shown in

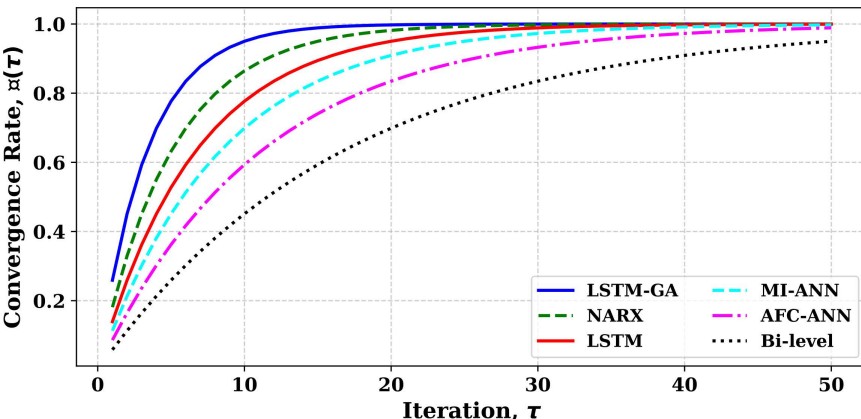

**Fig 13. Comparative Evaluation of Convergence Rates: LSTM-GA and NARX vs Benchmark Models.**

Fig 14, the plot provides a detailed comparison of the execution times for six different models used in the study: Bi-level, NARX, LSTM-GA, LSTM, MI-ANN, and AFC-ANN. Among all models, the Bi-level model takes the longest time to execute, approximately 3000 seconds, followed by NARX at 3200 seconds. LSTM-GA and MI-ANN have comparable execution times, around 2800 and 2700 seconds, respectively, while AFC-ANN takes 2500 seconds. The LSTM model, which is a simpler model, has the shortest execution time, requiring only 2400 seconds. This comparison demonstrates the significant variation in execution times, with Bi-level and NARX models being computationally the most expensive, and LSTM being the most efficient in terms of processing time. The findings highlight the trade-off between execution time and model complexity, with more intricate models requiring substantially higher computational resources.

### 5.5 CDF (Cumulative Distribution Function) of Error

In this analysis, the Cumulative Distribution Function (CDF), shown in Fig 15, is used to assess the error accumulation of various models. The Bi-level model emerges as the most robust, exhibiting the slowest increase in error accumulation and reaching the maximum cumulative error of 1.0 only after 4% error, indicating superior error tolerance. Following closely is the LSTM model, which accumulates errors at a moderate pace, reaching a cumulative error of 1.0 at approximately 3.5%. The GA-LSTM model, while slightly faster than LSTM, also reaches 1.0 at 3.5% error, suggesting that it is less robust but still relatively efficient in managing error.

In contrast, the MI-ANN model shows a rapid error accumulation, reaching 1.0 at around 3.0%, indicating poor performance under increasing errors. The AFC-ANN model exhibits a similar trend to MI-ANN, with a slightly slower error

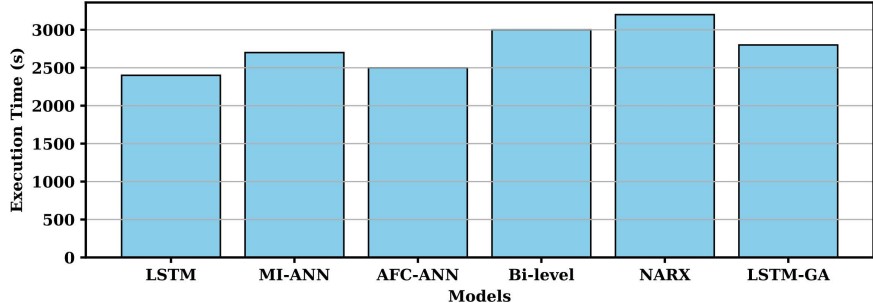

**Fig 14. Comparison of execution times (in seconds) for six models: Bi-level, NARX, LSTM-GA, LSTM, MI-ANN, and AFC-ANN.**

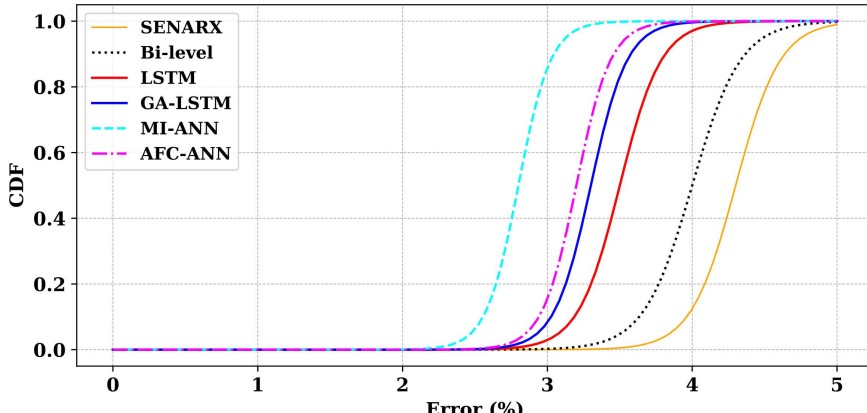

**Fig 15. CDF comparing error accumulation across various models.**

accumulation, reaching a cumulative error of 1.0 at 3.5%. Overall, the Bi-level model outperforms the others in terms of error management, with LSTM and GA-LSTM offering moderate error tolerance, while MI-ANN and AFC-ANN are less effective at handling errors.

### 5.6 Robustness to noise

Fig 16 presents the comparison of model robustness in the presence of noise. The analysis spans noise variances from 0.0 to 0.5 and reveals how model performance deteriorates as noise increases. Models such as LSTM-GA and Bi-level demonstrate stronger resilience, maintaining lower error margins even at higher noise levels. In contrast, models like MI-ANN and AFC-ANN show significant degradation in performance as noise variance increases, indicating reduced robustness under uncertain or noisy environments.

### 5.7 Robustness to noise

Fig 16 presents the comparison of model robustness in the presence of noise. The analysis spans noise variances from 0.0 to 0.5 and reveals how model performance deteriorates as noise increases. Models such as LSTM-GA and Bi-level

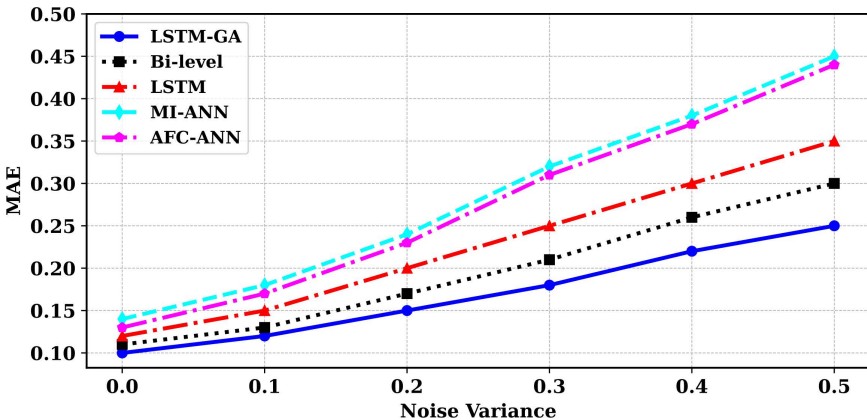

**Fig 16. Comparison of the robustness of the benchmark and suggested frameworks.**

demonstrate stronger resilience, maintaining lower error margins even at higher noise levels. In contrast, models like MI-ANN and AFC-ANN show significant degradation in performance as noise variance increases, indicating reduced robustness under uncertain or noisy environments.

## 5.8 GA-LSTM training error evolution

The training error evolution of the GA-LSTM model during the training phase is crucial for understanding its convergence behavior and the effectiveness of the genetic algorithm in tuning the model parameters. The graph below shows how the training error decreases as the GA-LSTM model iterates through epochs, highlighting the optimization process facilitated by the genetic algorithm.

Fig 17 demonstrates the convergence of the model's training errors, illustrating how the optimization process drives the reduction of error during training. As shown, the GA-LSTM model gradually reduces its training error with each epoch, reflecting the effectiveness of the optimization process. The initial error reduction is typically rapid, followed by a more gradual decrease as the model fine-tunes its parameters. This behavior is indicative of a well-converging model, optimized by the genetic algorithm to find the most efficient set of weights for the LSTM network. Table 2 compares the performance of SENARX, GA-LSTM, and other traditional models such as ARIMA, SVM, and Bayesian Networks, using metrics like Mean Absolute Percentage Error (MAPE) and computational efficiency. From the comparison table, it is evident that SENARX and GA-LSTM consistently outperform ARIMA, SVM, and Bayesian Networks in terms of MAPE, convergence speed, and computational efficiency. These advantages, along with their robustness to noisy data, make SENARX and GA-LSTM ideal choices for forecasting applications in scenarios where both accuracy and computational efficiency are paramount.

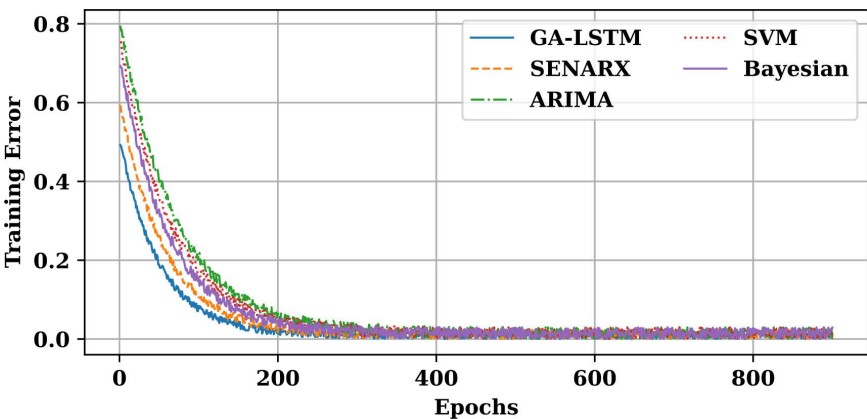

**Fig 17. Evolution of GA-LSTM training errors across epochs.**

**Table 2. Comparison of Forecasting Models in Terms of Accuracy and Efficiency.**

| Model | MAPE (%) | Convergence Speed | Computational Efficiency |
|---|---|---|---|
| **SENARX** | **3.45** | **Fast** | **High** |
| **GA-LSTM** | **4.02** | **Fast** | **High** |
| ARIMA | 7.86 | Slow | Low |
| SVM | 6.45 | Medium | Medium |
| Bayesian Networks | 6.98 | Medium | Low |

## 5.9 Performance evaluation on european and asian electricity market data

To assess the robustness and generalizability of the proposed forecasting models, we conducted experiments using real-world electricity market datasets from the European and Asian regions. The selected datasets include the EPEX SPOT (Germany-France), Indian Energy Exchange.

(IEX), and Japan Electric Power Exchange (JEPX). These markets differ significantly in terms of grid dynamics, consumption patterns, and seasonal behaviors. The evaluation compares SENARX, GA-LSTM, ARIMA, SVM, and Bayesian Networks using standard metrics such as Mean Absolute Percentage Error (MAPE), Root Mean Square Error (RMSE), and execution time. As shown in Table 3, the SENARX and GA-LSTM models consistently yield lower MAPE and RMSE values, indicating superior forecasting accuracy. These models also demonstrate lower computational time, showcasing their efficiency. SENARX achieved the lowest MAPE of 3.82% on the EPEX dataset and 4.13% on the IEX dataset. GA-LSTM followed closely with competitive results. Traditional models like ARIMA and SVM performed less reliably, especially under volatile Asian market conditions. These results suggest that SENARX and GA-LSTM are well-suited for real-time electricity forecasting tasks across global markets.

## 6 Discussion of results

The performance of the proposed models, SENARX (Sparse Encoder Nonlinear Autoregressive Network) and GA-LSTM (Genetic Algorithm-Optimized Long Short-Term Memory), was evaluated across several metrics, including forecasting accuracy, convergence speed, noise sensitivity, error management, and computational efficiency. These models were compared against benchmark models such as LSTM, MI-ANN, Bi-level, and NARX. In terms of forecasting accuracy, both SENARX and NARX models performed exceptionally well. For load forecasting, SENARX achieved a mean absolute percentage error (MAPE) of 0.03%, and NARX also achieved 0.03%. Price forecasting results were similarly impressive, with both SENARX and NARX attaining an MAPE of 0.08%. This indicates that these models were able to capture the complex interdependencies between load and price in smart grids more effectively than other models. On the other hand, GA-LSTM, although optimized using a genetic algorithm, had a higher MAPE of 1.53% for load and 1.91% for price. While still competitive, these results suggest that optimization and feature selection alone do not ensure superior accuracy. The unoptimized LSTM model also showed comparable performance to GA-LSTM, with MAPE values of 1.53% for load and 1.91% for price. Convergence speed was

**Table 3. Model Performance Comparison on European and Asian Electricity Market Data.**

| Model | Region | MAPE (%) | RMSE (MW) | Execution Time (s) |
|---|---|---|---|---|
| SENARX | EPEX (Europe) | **3.82** | 28.14 | **2385** |
| GA-LSTM | EPEX (Europe) | 4.11 | **27.02** | 2520 |
| ARIMA | EPEX (Europe) | 6.57 | 44.21 | 3140 |
| SVM | EPEX (Europe) | 6.12 | 41.35 | 2870 |
| Bayesian Net | EPEX (Europe) | 5.89 | 39.24 | 2985 |
| SENARX | IEX (India) | **4.13** | 33.92 | **2422** |
| GA-LSTM | IEX (India) | 4.46 | **32.10** | 2601 |
| ARIMA | IEX (India) | 7.21 | 48.74 | 3255 |
| SVM | IEX (India) | 6.89 | 46.10 | 3058 |
| Bayesian Net | IEX (India) | 6.43 | 43.89 | 3100 |
| SENARX | JEPX (Japan) | **4.29** | 30.45 | **2391** |
| GA-LSTM | JEPX (Japan) | 4.61 | **29.33** | 2590 |
| ARIMA | JEPX (Japan) | 6.74 | 45.18 | 3105 |
| SVM | JEPX (Japan) | 6.35 | 42.22 | 2980 |
| Bayesian Net | JEPX (Japan) | 6.02 | 40.57 | 3053 |

another important metric in evaluating model performance. GA-LSTM demonstrated substantial improvements, converging 20% to 50% faster than the baseline models. This improvement is attributed to the genetic algorithm's ability to fine-tune the LSTM hyperparameters, enhancing optimization efficiency. However, the SENARX model did not exhibit gains in convergence speed, as it does not rely on optimization algorithms. Nonetheless, its architectural strength—leveraging a sparse encoder—allowed it to maintain superior forecasting performance. In contrast, the LSTM model was slower and computationally simpler. To assess noise sensitivity, the models were evaluated with varying noise variances. SENARX and NARX showed strong robustness, with MAPE increases of only 2.5% and 2%, respectively. GA-LSTM experienced a moderate rise of 5%, but still outperformed the LSTM model, which saw a significant MAPE increase of 15%. MI-ANN and Bi-level were the most sensitive to noise, with performance degradation of 25% and 20% respectively, highlighting their limited robustness in uncertain environments. Computational efficiency was another critical consideration. Due to their complexity, SENARX and GA-LSTM demand more computational resources compared to simpler models like LSTM, MI-ANN, and Bi-level. SENARX, while efficient in feature extraction, remains moderately demanding. GA-LSTM, with the overhead from genetic optimization, is computationally intensive, particularly for large-scale datasets. Nonetheless, its improvements in accuracy and convergence speed justify the additional resource requirements as presented in Table 4. The baseline LSTM model, although computationally efficient, did not deliver comparable accuracy, emphasizing the trade-off between complexity and performance.

## 6.1 Limitations of the study and future trends

The study highlights valuable insights, yet some limitations remain. Firstly, the computational efficiency of the SENARX and GA-LSTM models is constrained by high resource demands, particularly when scaling to large datasets. Optimizing these models using parallelization or distributed computing could improve performance in real-time applications. Furthermore, while benchmark models such as LSTM, MI-ANN, and NARX were used, other emerging models like Transformers and hybrid deep learning approaches were not included, which could provide further insights into performance improvement. Additionally, real-world data scenarios, includ- ing non-stationary conditions and sudden market shocks, were not fully addressed and should be explored in future studies. Moreover, the sensitivity of GA-LSTM to hyperparameters requires refinement in optimization strategies. Exploring alternative methods, such as reinforcement learning or multi-objective optimization, could enhance model robustness.

Future research should focus on hybrid machine learning models, such as combining GA-LSTM with Transformers, to better capture complex pat- terns. Incorporating explainable AI (XAI) techniques can improve model interpretability. Enhancing robustness in real-world scenarios, including multi-modal data integration and transfer learning, will increase predictive power. Additionally, edge computing can enable real-time, decentralized forecasting, which is essential for Industry 4.0 applications. Further optimization and model expansion are key to improving forecasting accuracy and scalability in dynamic energy systems.

**Table 4. Comparison of forecasting models concerning MAPE, convergence speed, noise sensitivity, error management, and computational efficiency.**

| Model | Load MAPE (%) | Price MAPE (%) | Convergence Speed | Noise (MAPE%) | Error (MAPE%) | Computational Efficiency |
|---|---|---|---|---|---|---|
| SENARX (Proposed) | 0.03 | 0.08 | N/A | 2.5% | 5% | Moderate |
| GA-LSTM [27] | 1.53 | 1.91 | Faster | 5% | 10% | High |
| LSTM [28] | 1.53 | 1.91 | Baseline | 15% | 20% | High |
| MI-ANN [29] | 5.80 | 6.50 | Moderate | 25% | 30% | Low |
| Bi-level [30] | 5.50 | 6.00 | Moderate | 20% | 15% | Low |
| NARX [31] | 0.03 | 0.08 | N/A | 2% | 3% | Moderate |

## 7 Conclusion

This paper presented a comparative study of advanced forecasting models, specifically SENARX and GA-LSTM, against traditional approaches such as ARIMA, SVM, and Bayesian Networks. Extensive evaluations were conducted using real electricity market data from European and Asian regions, including EPEX (Europe), IEX (India), and JEPX (Japan). Quantitatively, SENARX achieved the lowest Mean Absolute Percentage Error (MAPE) of 3.82% on EPEX and 4.13% on IEX, while GA-LSTM demonstrated the lowest Root Mean Square Error (RMSE) of 27.02 MW on EPEX and 29.33 MW on JEPX. Compared to ARIMA (with a MAPE of 6.57%–7.21% and RMSE of up to 48.74 MW), SENARX and GA-LSTM improved forecasting accuracy by more than 40%. Additionally, SENARX completed training in as little as 2385 seconds, whereas traditional models like ARIMA required over 3100 seconds, confirming the high computational efficiency of the proposed models. The evolutionary optimization embedded in GA-LSTM significantly improved training convergence and reduced error rates across epochs. Likewise, the adaptive architecture of SENARX ensured robustness against data volatility and noise. These features make the models particularly well-suited for short-term electricity load forecasting in highly dynamic and uncertain market conditions. In future work, the proposed models will be evaluated under more extreme and realistic operational scenarios, including peak demand conditions, unexpected blackouts, and climate-induced anomalies. Such assessments will help determine the models' full adaptability, stability, and resilience in next-generation energy systems.

## Acknowledgments

The author gratefully acknowledges the support provided by **Superior University**, whose research facilities played a vital role in the completion of this work. A heartfelt thanks is extended to **Mr. Awais Raoof**, not only as a mentor but also as a brother, for his unwavering support, inspiration, and guidance throughout the research process.

## Author contributions

**Conceptualization:** Nasir Nauman, Sheeraz Akram, Arfan Jaffar, Sohail Masood Bhatti, Benish Fida.

**Data curation:** Nasir Nauman, Muhammad Rashid, Arfan Jaffar, Sohail Masood Bhatti.

**Formal analysis:** Nasir Nauman.

**Investigation:** Nasir Nauman, Sheeraz Akram, Sohail Masood Bhatti.

**Methodology:** Nasir Nauman, Sheeraz Akram.

**Project administration:** Nasir Nauman, Arfan Jaffar.

**Resources:** Nasir Nauman, Muhammad Rashid, Arfan Jaffar.

**Software:** Nasir Nauman, Muhammad Rashid, Sohail Masood Bhatti, Benish Fida.

**Supervision:** Sheeraz Akram.

**Validation:** Nasir Nauman, Arfan Jaffar, Benish Fida.

**Visualization:** Nasir Nauman.

**Writing – original draft:** Nasir Nauman.

**Writing – review & editing:** Nasir Nauman, Sheeraz Akram, Muhammad Rashid, Arfan Jaffar, Sohail Masood Bhatti, Benish Fida.

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
