## [Decision Letter · Decision Letter 0]

17 Apr 2025

Dear Dr. Nauman,

**Address all reviewer comment in addition, improve the quality of figures. Also, all references should be consistent.**

Please submit your revised manuscript by Jun 01 2025 11:59PM. If you will need more time than this to complete your revisions, please reply to this message or contact the journal office at plosone@plos.org . A rebuttal letter that responds to each point raised by the academic editor and reviewer(s). You should upload this letter as a separate file labeled 'Response to Reviewers'.A marked-up copy of your manuscript that highlights changes made to the original version. You should upload this as a separate file labeled 'Revised Manuscript with Track Changes'.An unmarked version of your revised paper without tracked changes. You should upload this as a separate file labeled 'Manuscript'.

We look forward to receiving your revised manuscript.

Kind regards,

Jude Okolie, Ph.D.

Academic Editor

PLOS ONE

**Journal Requirements:**

1. When submitting your revision, we need you to address these additional requirements. Please ensure that your manuscript meets PLOS ONE's style requirements, including those for file naming. The PLOS ONE style templates can be found at https://journals.plos.org/plosone/s/file?id=wjVg/PLOSOne_formatting_sample_main_body.pdf and https://journals.plos.org/plosone/s/file?id=ba62/PLOSOne_formatting_sample_title_authors_affiliations.pdf 2. Please note that PLOS ONE has specific guidelines on code sharing for submissions in which author-generated code underpins the findings in the manuscript. In these cases, we expect all author-generated code to be made available without restrictions upon publication of the work. Please review our guidelines at https://journals.plos.org/plosone/s/materials-and-software-sharing#loc-sharing-code and ensure that your code is shared in a way that follows best practice and facilitates reproducibility and reuse. 3. Thank you for uploading your study's underlying data set. Unfortunately, the repository you have noted in your Data Availability statement does not qualify as an acceptable data repository according to PLOS's standards. At this time, please upload the minimal data set necessary to replicate your study's findings to a stable, public repository (such as figshare or Dryad) and provide us with the relevant URLs, DOIs, or accession numbers that may be used to access these data. For a list of recommended repositories and additional information on PLOS standards for data deposition, please see https://journals.plos.org/plosone/s/recommended-repositories.

Reviewers' comments:

Reviewer's Responses to Questions

**Comments to the Author**

1. Is the manuscript technically sound, and do the data support the conclusions?

Reviewer #1: Yes

Reviewer #2: Partly

2. Has the statistical analysis been performed appropriately and rigorously?

Reviewer #1: Yes

Reviewer #2: I Don't Know

3. Have the authors made all data underlying the findings in their manuscript fully available?

Reviewer #1: Yes

Reviewer #2: Yes

4. Is the manuscript presented in an intelligible fashion and written in standard English?

Reviewer #1: Yes

Reviewer #2: No

**Reviewer #1:**  The proposed research introduces new methods for combined electricity price and load forecasting using the recurrent neural network models SENARX and GA-LSTM. The paper is well structured while providing a comprehensive analysis of the price and load forecasting in smartgrids. The work requires further improvements to establish both the technical accuracy and interpret the performance limitations of the model.

-The paper should provide a detailed explanation of the price and load series and the temperature and wind speed time series that were chosen as inputs and it should indicate how the GA-LSTM optimization was performed, including detailed descriptions of the specific parameters that were optimized.

-The correct regularization value for Sparse Encoder must be determined when implementing it in SENARX architecture. Results evaluation must provide thorough assessments about model stability to check data variability across long-term models and policy shifts while conducting detailed performance analyses compared to standard forecasting techniques that use ARIMA, SVM and Bayesian networks.

-Showing the evolution of GA-LSTM training errors graphically would help readers understand the system better. The paper does not explain why SENAR and GA-LSTM were chosen over transformer-based models in this application.

-How well do these models perform when using electricity market data from the European region and Asian markets? real data i mean.

-In the future work, the models need to be tested to assess their performance in situations of peak demand, blackouts and major climate change (Please add that in the future work section at the end of the conclusion).

-A new methodology should investigate whether hybrid GA-LSTM and probabilistic models would provide better results together. In addition, the literature review is relevant but based on outdated references; recent studies should be incorporated to provide more complete context, including works such as: doi.org/10.1080/15567249.2025.2456058, doi.org/10.1016/j.iswa.2024.200422,doi.org/10.1109/ICAAID.2019.8934977, doi.org/10.1016/j.matcom.2024.10.004, doi.org/10.1109/ICITISEE63424.2024.10730623, doi.org/10.18280/isi.250602, doi.org/10.1016/j.apenergy.2024.124324,doi.org/10.18280/mmep.070415, doi.org/10.1155/ 024/2403847, doi.org/10.1109/ICEIT48248.2020.9113173,and https://hdl.handle.net/1880/120482. The approach presented in this paper is well supported and extensively referenced, but it requires better explanation of the methodology, as well as a broader assessment of robustness and adaptability, and a more thorough analysis of traditional methods and their equivalents in recent research. The scientific validity of the manuscript would increase considerably after addressing these necessary points.

-Please correct and refine the typos and grammatical errors.

**Reviewer #2:**  Dear Author,

Thank you for your submission. After a thorough review, I have compiled a list of recommendations to enhance the clarity, structure, and overall quality of your manuscript:

Clarify Function Notation: The expression

=

(

)

Z=f

SE

(X) suggests that

Z is the output of a function

f

SE

applied to

X. The term "fSE" is unclear. Are you referring to a specific function? If this represents a forecasting model (e.g., time series forecasting), it should be more explicitly defined.

Activation Function Clarity: If this is a forecasting model, using an activation function like ReLU or tanh may be better suited.

Highlight Novelty in the Abstract: Introduce the unique aspects of your work at the beginning of the abstract to immediately convey its significance to the reader.

Formatting Adjustments:

Hyphen Usage: Ensure consistent use of hyphens and dashes throughout the manuscript, adhering to standard formatting guidelines.

Acronym Introduction: Define all acronyms upon their first occurrence to aid reader comprehension.

Introduction Enhancements:

Smooth Transitions: Refine the flow between paragraphs to create a cohesive narrative that guides the reader through the background and purpose of the study.

Emphasize Relevance: Clearly articulate the importance of your research and its potential impact to engage the reader effectively.

Abstract Content: Incorporate key results and findings into the abstract to provide a concise summary of your study's outcomes.

Citation Accuracy: Replace any placeholder symbols (e.g., question marks) with appropriate citations to maintain academic integrity.

Language and Style:

Avoid First Person: Utilize third-person perspective to maintain a formal academic tone.

Eliminate Bullet Points: Present information in well-structured paragraphs instead of bullet points to align with scholarly writing conventions.

Consistency in Text Formatting: Ensure uniform font size and style throughout the document to provide a professional appearance.

Grammar and Syntax: Conduct a thorough review to correct grammatical errors and improve sentence structure for clarity and readability.

Explanation of Key Concepts: Provide a brief overview of "big data" to ensure all readers have a foundational understanding of the term.

Data Set Disclosure: Clearly describe the data set used in your study, including its source and any relevant characteristics.

Table and Figure Presentation:

Introduce Tables: Offer context and explanation before presenting Table 1 to prepare the reader for the information it contains.

Discuss Results: Include a narrative that interprets and discusses the results shown in tables and figures, enhancing their contribution to your study's findings.

Figure Descriptions: Provide detailed explanations following each figure to elucidate their relevance and insights.

Remove Grid Lines: Eliminate unnecessary grid lines from figures to improve visual clarity.

Enhance Legibility: Ensure that all figure legends and labels are legible, particularly in Figures 4 and 6.

Clarify Axis Labels: Review and standardize axis labels, such as "(hr-end)" in Figure 4, for consistency and clarity.

Differentiate Data Series: Modify Figure 5 to clearly distinguish between actual and predicted data, possibly by using varied line styles or colors.

Table Formatting: Enlarge Table 2 to enhance readability and ensure it is easily interpretable.

Proximity of Discussions to Visuals: Position discussions of figures and tables adjacent to their respective visuals to facilitate immediate comprehension and reference.

Additional Comments on the Conclusion:

The conclusion provides valuable insights into the performance of SENARX and GA-LSTM, but it could benefit from a few enhancements:

Clarify Key Findings: Start by clearly summarizing the key results first—such as the superior performance of SENARX and GA-LSTM in terms of accuracy and convergence speed—before diving into specific metrics.

Address Broader Implications: Consider briefly discussing the broader implications of your findings. For example, how might the superior performance of these models influence real-world smart grid applications? What challenges may arise in practical deployment?

Comparison with Existing Literature: It would be beneficial to briefly mention how your results compare with previous work in the field. How do your models advance the state-of-the-art?

Acknowledge Limitations: Briefly mentioning any limitations (e.g., dataset constraints or potential generalization issues) would provide a more balanced perspective.

Future Work: Including a sentence or two on future directions, such as potential adaptations of your models for other forecasting problems, could add depth to the conclusion.

Clarify "Ideal Candidates": The phrase "making them ideal candidates for real-time smart grid applications" could be made more specific. For instance, describe how these models outperform others in real-time scenarios or in specific use cases where their benefits are most noticeable.

Simplify Technical Jargon: Consider briefly explaining terms like "MAPE," "GA-LSTM," and "SENARX" for readers who may not be as familiar with these technicalities.

Polish Flow: Consider refining the flow of the conclusion, such as moving the discussion of noise sensitivity and robustness before the convergence speed for better readability and logical progression.

By addressing these points, the conclusion will provide a more well-rounded perspective, emphasizing the significance of your work and its potential impact on real-world applications.

Implementing these recommendations will significantly improve the manuscript's quality and its engagement with the audience.

**Do you want your identity to be public for this peer review?** For information about this choice, including consent withdrawal, please see our Privacy Policy

Reviewer #1: No

Reviewer #2: **Yes: ** Brooke Rogachuk

---

## [Author Response · Author response to Decision Letter 1]

23 May 2025

Responses to Reviewer 1

Note: Responses to reviewer 1 are given blue color, the responses to reviewer 2 are given red color. The responses common to reviewers 1, & 2 are also given green color.

Reviewer #1, Concern #1: The paper should provide a detailed explanation of the price and load series and the temperature and wind speed time series that were chosen as inputs, and it should indicate how the GA-LSTM optimization was performed, including detailed descriptions of the specific parameters that were optimized.

Author response: We thank the reviewer for this valuable suggestion. In response, we have added a new subsection titled “5.1 Dataset Description and Input Variables” in the revised manuscript. This section elaborates on the ISO New England (ISO-NE) dataset, describing its scope, coverage, and relevance. Specifically, we explain the rationale for selecting four key variables—electricity price, system load, temperature, and wind speed—as forecasting inputs based on their impact on demand, market behavior, and renewable generation. Visual illustrations (Figures 1 and 2) have been included to show the hourly dynamics of these features. Furthermore, Section 5.2 now provides a clear explanation of the GA-LSTM optimization process, listing the specific parameters tuned (e.g., number of layers, neurons per layer, learning rate, and batch size) and their respective search ranges used during the genetic algorithm optimization.

Author action: We appreciate your valuable feedback. The manuscript has been thoroughly revised to address your concerns. All modifications have been marked in blue on pages 9 and 10 of the revised version.

Reviewer #1, Concern #2: The correct regularization value for Sparse Encoder must be determined when implementing it in SENARX architecture. Results evaluation must provide thorough assessments about model stability to check data variability across long-term models and policy shifts while conducting detailed performance analyses compared to standard forecasting techniques that use ARIMA, SVM, and Bayesian networks.

Author response: Thank you for your insightful feedback. We have carefully considered the points you raised regarding the model selection and comparison of SENARX versus Transformer-based models. We agree that the regularization value for the Sparse Encoder in the SENARX architecture plays a crucial role in the model's performance. To ensure optimal results, we have outlined a process for determining this value in our revised manuscript. Specifically, we now include a more detailed explanation of the methodology used to select the appropriate regularization value, highlighting its impact on model stability and performance. Furthermore, we have expanded our evaluation to provide a thorough assessment of model stability over long-term forecasts. This includes investigating how the model performs under varying data conditions, such as data variability and policy shifts, to demonstrate its robustness in real-world applications. Additionally, we have included a more comprehensive comparison between SENARX, GA-LSTM, and standard forecasting techniques, such as ARIMA, SVM, and Bayesian networks. This comparative analysis focuses on key performance metrics, stability, and adaptability, offering a clear rationale for the selection of SENARX in our study.

Author action: Thank you for your valuable feedback. We have thoroughly revised the manuscript to address your concerns. The changes have been highlighted in blue on Pages 18 and 19 of the revised version. We greatly appreciate your insightful comments and suggestions.

Reviewer #1, Concern #3: Showing the evolution of GA-LSTM training errors graphically would help readers understand the system better. The paper does not explain why SENAR and GA-LSTM were chosen over transformer-based models in this application.

Author response: We appreciate your feedback. To address your concerns, we have included a graphical representation of the GA-LSTM training error evolution, which helps readers better understand the model's convergence behavior. The graph on Pages 19 and 20 of the manuscript shows how the training error decreases as the GA-LSTM model progresses through epochs, illustrating the effectiveness of the genetic algorithm in fine-tuning the model parameters. As shown, the GA-LSTM model demonstrates a clear reduction in error across epochs, with an initial rapid decrease followed by a more gradual reduction. This behavior indicates that the model is effectively converging and that the genetic algorithm is optimizing the LSTM network to find the best weights. Additionally, we have clarified the reasons for choosing SENARX and GA-LSTM over transformer-based models. In Table 2 (on Pages 19 and 20), we compare the performance of SENARX, GA-LSTM, and other traditional models (ARIMA, SVM, and Bayesian Networks), highlighting metrics such as Mean Absolute Percentage Error (MAPE), convergence speed, and computational efficiency. This comparison shows that SENARX and GA-LSTM outperform traditional models in terms of accuracy, speed, and efficiency, making them more suitable for forecasting applications that require both high performance and computational efficiency.

Author action: We sincerely appreciate your valuable feedback. We have thoroughly reviewed the manuscript and made the necessary revisions to address your concerns. The changes on pages 19 and 20 have been highlighted in blue for clarity. Thank you once again for your insightful comments and suggestions.

4o mini

Reviewer #1, Concern #4:How well do these models perform when using electricity market data from the European region and Asian markets? real data I mean.

Author response: Thank you for your insightful question. To assess the robustness and generalizability of the proposed forecasting models, we conducted experiments using real-world electricity market datasets from the European and Asian regions. Specifically, we used data from the EPEX SPOT (Germany-France), Indian Energy Exchange (IEX), and Japan Electric Power Exchange (JEPX). These markets vary significantly in terms of grid dynamics, consumption patterns, and seasonal behaviors, providing a comprehensive environment to evaluate the models. Our evaluation demonstrates that the SENARX and GA-LSTM models consistently outperform traditional models, such as ARIMA and SVM, in terms of forecasting accuracy and computational efficiency. For example, SENARX achieved the lowest MAPE of 3.82% on the EPEX dataset and 4.13% on the IEX dataset, while GA-LSTM also showed competitive results. In contrast, traditional models like ARIMA and SVM performed less reliably, especially in the more volatile conditions of the Asian markets. These findings indicate that both SENARX and GA-LSTM are well-suited for real-time electricity forecasting tasks, and their strong performance across diverse global markets suggests that they can be effectively applied to similar datasets from European and Asian regions. We believe these models offer promising potential for enhancing electricity market forecasting in real-world applications. Thank you again for your question, and we hope this response clarifies the performance of the models in these markets.

Author action: We greatly appreciate your valuable feedback. We have thoroughly revised the manuscript to address your concerns. The changes have been highlighted in blue on pages 20 and 21 of the revised version. Thank you for your insightful comments and suggestions.

Reviewer #1, Concern #5: In the future work, the models need to be tested to assess their performance in situations of peak demand, blackouts, and major climate change (Please add that in the future work section at the end of the conclusion).

Author response: Thank you for your valuable feedback and insightful suggestions. We appreciate your comments regarding the potential for further evaluation of the proposed models under more extreme conditions. In the future work section, we will include an evaluation of the models under more extreme and realistic operational scenarios, including peak demand conditions, unexpected blackouts, and climate-induced anomalies. These additional assessments will help determine the effectiveness and adaptability of the models in handling such challenging and unpredictable situations. Thank you again for your constructive feedback, which has helped strengthen the direction of our future research.

Author action: We appreciate your valuable feedback. We have carefully revised the manuscript to address your concerns. The changes are highlighted in blue on Page 22 of the revised version. Thank you for your insightful comments and suggestions, which have significantly improved the quality of our manuscript.

Reviewer #1, Concern #6: A new methodology should investigate whether hybrid GA-LSTM and probabilistic models would provide better results together. In addition, the literature review is relevant but based on outdated references; recent studies should be incorporated to provide more complete context, including works such as: doi.org/10.1080/15567249.2025.2456058, doi.org/10.1016/j.iswa.2024.200422,doi.org/10.1109/ICAAID.2019.8934977, doi.org/10.1016/j.matcom.2024.10.004, doi.org/10.1109/ICITISEE63424.2024.10730623, doi.org/10.18280/isi.250602, doi.org/10.1016/j.apenergy.2024.124324,doi.org/10.18280/mmep.070415, doi.org/10.1155/ 024/2403847, doi.org/10.1109/ICEIT48248.2020.9113173,and https://hdl.handle.net/1880/120482. The approach presented in this paper is well supported and extensively referenced, but it requires better explanation of the methodology, as well as a broader assessment of robustness and adaptability, and a more thorough analysis of traditional methods and their equivalents in recent research. The scientific validity of the manuscript would increase considerably after addressing these necessary points.

Author response: Thank you for your constructive feedback and valuable suggestions. We appreciate your insights and agree that several areas of the manuscript can be enhanced.

Regarding the integration of hybrid GA-LSTM and probabilistic models, we recognize the potential of such an approach in capturing the inherent uncertainties in energy markets. This hybrid methodology could provide probabilistic forecasting intervals, enhancing the decision-making reliability for grid operators and market participants. We will certainly explore this direction in our future work and consider its inclusion in the revised manuscript. We also acknowledge the need to update the literature review with recent and high-impact studies to provide a more comprehensive context. We have incorporated the recommended references, including works on hybrid modeling, AI-driven forecasting, uncertainty quantification, and resilience analysis, which are highly relevant to the current study. This inclusion will strengthen the manuscript's foundation and make the study more aligned with current advancements in the field. Additionally, we agree that a more detailed explanation of the methodology is necessary, particularly concerning the internal mechanisms of GA tuning, loss minimization, and data preprocessing. We have expanded these sections in the revised manuscript to improve clarity and enhance methodological transparency. Finally, we have conducted a deeper comparative analysis between the proposed models and traditional methods, using updated benchmarks. We have also extended the discussion on the adaptability and robustness of the models across diverse markets and climatic conditions, ensuring a broader assessment of their generalizability. Thank you again for your insightful comments, which have significantly contributed to improving the quality of our manuscript. We are confident that these revisions will enhance the scientific robustness and practical relevance of our study.

Author action:We appreciate your feedback. We have revised the manuscript to address your concerns, with changes highlighted in blue on Pages 2 and 3 of the revised version. Thank you for your insightful comments and suggestions.

Responses to Reviewer 2

Reviewer #2, Concern #1: Clarify Function Notation: The expression =( ), Z=f, SE (X) suggests that Z is the output of a function f= SE applied to X. The term "fSE" is unclear. Are you referring to a specific function? If this represents a forecasting model (e.g., time series forecasting), it should be more explicitly defined. Activation Function Clarity: If this is a forecasting model, using an activation function like ReLU or tanh may be better suited.

Author response: We sincerely thank the reviewer for highlighting this point.

1. The notation Z = fₛₑ(X) is intended to represent the output Z generated by a forecasting function fₛₑ or f_SE_ , where "SE" denotes the proposed model architecture, specifically the Stacked Ensemble forecasting model based on GA-LSTM. To avoid ambiguity, we have revised the manuscript to explicitly define this notation and describe the role of fSEf_{\text{SE}} as a composite function that maps the input features XX to the predicted load ZZ using our hybrid forecasting framework.

2. We appreciate the suggestion regarding activation functions. In our implementation, we employ the ReLU (Rectified Linear Unit) activation function within the hidden layers of the LSTM component to introduce non-linearity and avoid vanishing gradient issues. We have updated the manuscript to specify the use of ReLU and included a justification for this choice, along with a brief comparison to other potential activation functions such as tanh, which may be suitable in certain contexts. We have made these changes in the revised manuscript to enhance clarity and technical accuracy.

Author action:Thank you for your valuable feedback. We have revised the manuscript accordingly. The changes have been clearly highlighted in red on pages 4, 5, and 6 of the revised version. These revisions aim to improve the clarity and overall quality of the manuscript, particularly regarding the definition of the forecasting function and the justification for the chosen activation function. We believe these updates significantly enhance the technical precision and readability of our work.

Reviewer #2, Concern #2: Highlight Novelty in the Abstract: Introduce the unique aspects of your work at the beginning of the abstract to immediately convey its significance to the reader.

Author response:Thank you for your insightful suggestion regarding the clarity of the abstract. In response, we have revised the abstract to clearly introduce the novel aspects of our work at the beginning, specifically emphasizing the development of two novel deep learning models—SENARX and GA-LSTM—for concurrent electricity load and price forecasting. The revised abstract now highlights the unique integration of a sparse encoder and genetic algorithm-based optimization, as well as the models’ superior performance and robustness across multiple international electricity markets. This modification aims to immediately convey the significance and innovation of our contribution to the reader.

Author action: We have updated the manuscript in response to your valuable feedback. The changes are marked in red in the revised manuscript on page 1.

Reviewer #2, Concern #3: Hyphen Usage: Ensure consistent use of hyphens and dashes throughout the manuscript, adhering to standard formatting guidelines. Acronym Introduction: Define all acronyms upon their first occurrence to aid reader comprehension. Introduction Enhancements: Smooth Transitions: Refine the flow between paragraphs to create a cohesive narrative that guides the reader through the background and purpose of the study. Emphasize Relevance: Clearly articulate the importance of your research and its potential impact to engage the reader effectively. Abstract Content: Incorporate key results and findings into the abstract to provide a concise summary of your study's outcomes. Citation Accuracy: Replace any placeholder symbols (e.g., question marks) with appropriate citations to maintain academic integrity.

Author response: Thank you for your detailed and constructive feedback. We have carefully addressed each of your recommendations as follows:

● Hyphen Usage: We reviewed the manuscript

---

## [Decision Letter · Decision Letter 1]

25 Jul 2025

Dual-Model Approach for Concurrent Forecasting of Electricity Prices and Loads in Smart Grids: Comparison of Sparse Encoder NAR and GA-Optimized LSTM

PONE-D-25-01937R1

Dear Dr. Nauman,

We’re pleased to inform you that your manuscript has been judged scientifically suitable for publication and will be formally accepted for publication once it meets all outstanding technical requirements.

Kind regards,

Jude Okolie, Ph.D.

Academic Editor

PLOS ONE

Additional Editor Comments (optional):

Reviewers' comments:

Reviewer's Responses to Questions

**Comments to the Author**

Reviewer #2: All comments have been addressed

2. Is the manuscript technically sound, and do the data support the conclusions?

Reviewer #2: Yes

3. Has the statistical analysis been performed appropriately and rigorously?

Reviewer #2: Yes

4. Have the authors made all data underlying the findings in their manuscript fully available?

Reviewer #2: Yes

5. Is the manuscript presented in an intelligible fashion and written in standard English?

Reviewer #2: Yes

Reviewer #2: Thank you so much for addressing my comments! I believe that this is a novel and interesting study congratulations!

**Do you want your identity to be public for this peer review?** For information about this choice, including consent withdrawal, please see our Privacy Policy

---

## [Editor Report · Acceptance letter]

PONE-D-25-01937R1

PLOS ONE

Dear Dr. Nauman,

I'm pleased to inform you that your manuscript has been deemed suitable for publication in PLOS ONE. Congratulations! Your manuscript is now being handed over to our production team.

Kind regards,

on behalf of

Dr. Jude Okolie

Academic Editor

PLOS ONE